# GDR-GMA: Machine Unlearning via Direction-Rectified and Magnitude-Adjusted Gradients

Shen Lin
State Key Laboratory of Integrated
Service Networks (ISN)
Xidian University
Xi'an, China
linshen@stu.xidian.edu.cn

Xiaoyu Zhang
State Key Laboratory of Integrated
Service Networks (ISN)
Xidian University
Xi'an, China
xiaoyuzhang@xidian.edu.cn

Willy Susilo
School of Computing and Information
Technology
University of Wollongong
Wollongong, Australia
wsusilo@uow.edu.au

Xiaofeng Chen*
State Key Laboratory of Integrated
Service Networks (ISN)
Xidian University
Xi'an, China
xfchen@xidian.edu.cn

Jun Liu
Lancaster University
Lancaster, United Kingdom
Singapore University of Technology
and Design
Singapore, Singapore
j.liu81@lancaster.ac.uk

## Abstract

As concerns over privacy protection grow and relevant laws come into effect, machine unlearning (MU) has emerged as a pivotal research area. Due to the complexity of the forgetting data distribution, the sample-wise MU is still open challenges. Gradient ascent, as the inverse of gradient descent, is naturally applied to machine unlearning, which is also the inverse process of machine learning. However, the straightforward gradient ascent MU method suffers from the trade-off between effectiveness, fidelity, and efficiency. In this work, we analyze the gradient ascent MU process from a multi-task learning (MTL) view. This perspective reveals two problems that cause the trade-off, *i.e.,* the gradient direction problem and the gradient dominant problem. To address these problems, we propose a novel MU method, namely GDR-GMA, consisting of Gradient Direction Rectification (GDR) and Gradient Magnitude Adjustment (GMA). For the gradient direction problem, GDR rectifies the direction between the conflicting gradients by projecting a gradient onto the orthonormal plane of the conflicting gradient. For the gradient dominant problem, GMA dynamically adjusts the magnitude of the update gradients by assigning the dynamic magnitude weight parameter to the update gradients. Furthermore, we evaluate GDR-GMA against several baseline methods in three sample-wise MU scenarios: random data forgetting, sub-class forgetting, and class forgetting. Extensive experimental results demonstrate the superior performance of GDR-GMA in effectiveness, fidelity, and efficiency. Code is available at https://github.com/RUIYUN-ML/GDR-GMA.

*Corresponding author

## CCS Concepts

• **Security and privacy → Privacy protections**.

## Keywords

Deep learning, Machine unlearning, Gradient

**ACM Reference Format:**
Shen Lin, Xiaoyu Zhang, Willy Susilo, Xiaofeng Chen, and Jun Liu. 2024. GDR-GMA: Machine Unlearning via Direction-Rectified and Magnitude-Adjusted Gradients. In *Proceedings of the 32nd ACM International Conference on Multimedia (MM '24), October 28-November 1, 2024, Melbourne, VIC, Australia.* ACM, New York, NY, USA, 9 pages. https://doi.org/10.1145/3664647.3680775

## 1 Introduction

With the widespread use of artificial intelligence (AI) techniques [2, 12, 13, 16, 29, 37, 40], concerns about privacy protection have escalated. To address these concerns, an increasing number of regulations and laws have been introduced on privacy protection, such as the European Union's GDPR (General Data Protection Regulation) [34]. The GDPR has been promulgated to give people the *right-to-be-forgotten*, which requires information service providers to delete personal data on request from the data owner. Furthermore, this regulation stipulates that providers must also erase the corresponding influence of the requested data. Numerous studies [15, 18, 19, 28, 30, 31, 38, 39] have demonstrated that machine learning models possess the ability to memorize data samples. For example, membership inference attacks [15, 30, 31] can infer whether a data sample is in the training set or not. Consequently, AI service providers must remove the requested data samples and their associated memorized knowledge from the models. However, the naive approach of Retraining the model from Scratch after removing the forgetting data samples from the training set is prohibitively expensive in practice. Therefore, a new research direction for privacy protection emerged to efficiently remove the knowledge of requested data samples, called *machine unlearning (MU)*.

The sample-wise unlearning methods focus on unlearning a subset of data samples in the training set, which can be random data samples, a sub-class of data samples, and a class of data samples. Therefore, it is challenging for the sample-wise MU methods to handle the complex distribution of forgetting data samples. Many works [1, 3, 5, 9–11, 14, 17, 22, 23, 32, 35] attempted to address the sample-wise MU challenges. For example, Bourtoule et al. [1] proposed an exact unlearning method, SISA, to unlearn data samples by retraining the sub-models. However, SISA needs to retrain plenty of sub-models when the requested target data samples are widely distributed across the different shards. To further improve efficiency, many approximate unlearning methods [1, 5, 7, 9–11, 14, 22, 24, 33] were proposed. They estimate the contribution of the forgetting data samples and unlearn them by updating model parameters. Specially, gradient plays an important role in these cutting-edge works. For example, Graves et al. [10] updated the model with the relevant stored gradients. Fan et al. [7] used the ascent gradient to generate the weight salience map.

As one of the basic methods, the straightforward gradient ascent MU method still inspires state-of-the-art works [10, 14, 22, 30, 33], namely the negative gradient (NegGrad). It can be seen as a multi-task learning process: unlearning the forgetting dataset and maintaining the remaining dataset. For image classification, it calculates the ascent gradient of the forgetting data samples and the descent gradient of the remaining data samples to perform a joint model update. Although NegGrad can misclassify forgetting data samples with few epochs, it will also misclassify most of the remaining samples, leading to a sharp decline in the classification performance (low fidelity). In contrast, the forgetting samples are barely unlearned if NegGrad maintains the classification performance, resulting in MU's low effectiveness and efficiency. Therefore, it suffers from the trade-off between effectiveness, fidelity, and efficiency.

In this work, we identify two key factors for this trade-off problem, *i.e.,* **gradient direction problem** and **gradient domination problem**, as shown in Sec. 3.3 and Sec. 3.4. For the gradient direction problem, we identify two pairs of direction conflicts: 1) the ascent gradient of the forgetting data samples and the descent gradient of the remaining data samples; 2) the descent gradient of the remaining data samples and the ascent gradient of the other forgetting data samples (from the difference set of the current forgetting samples and the forgetting set). For the gradient domination problem, we find that a single gradient always dominates the update process, so the MU process solely tends to one task.

To address the above problems and to better apply the gradient ascent to the sample-wise machine unlearning task, we propose a novel unlearning method consisting of Gradient Direction Rectification (GDR) and Gradient Magnitude Adjustment (GMA), dubbed GDR-GMA. For the gradient direction problem, GDR rectifies the direction of gradients to be orthogonal to the conflict gradients. For the gradient domination problem, GMA dynamically adjusts the magnitude of update gradients to maintain a balance state of two tasks in model updating. We summarize our contributions to this paper as follows:

- We analyze the problems in the straightforward gradient ascent MU method that causes the trade-off among effectiveness, fidelity, and efficiency from a multi-task learning

view. This perspective reveals two problems: the gradient direction problem and the gradient domination problem.
- We propose the GDR-GMA unlearning method to address these two problems. For the gradient direction problem, GDR-GMA rectifies the direction between the conflicting gradients by projecting a gradient onto the orthonormal plane of the conflicting gradient. For the gradient domination problem, GDR-GMA dynamically adjusts the magnitude of the update gradients by the dynamic magnitude weight parameters.
- We conduct extensive experiments in three sample-wise MU scenarios: random data forgetting, sub-class forgetting, and class forgetting. Compared to 11 baseline MU methods, GDR-GMA achieves a superior performance in effectiveness, efficiency, and fidelity.

## 2 Related Work

**Multi-task learning.** Multi-task learning (MTL) aims to help improve the model performance by leveraging the commonalities and differences across multiple tasks. Instead of training on a single task, the model is trained simultaneously on multiple related tasks. In MTL, reducing the direction conflict and domination by a single task is an important topic. For example, Chen et al. [4] normalized the gradient to adaptive balance the loss among tasks. Yu et al. [36] pointed out that the gradient direction conflict and single task domination may damage the model performance. Liu et al. [25] proposed a Multi-Task Attention Network to improve performance. Although MU is not a traditional MTL task, we can utilize the philosophy behind MTL to mitigate the gradient direction and dominant problems in MU.

**Class-wise machine unlearning.** As a particular case of the sample-wise MU, the class-wise machine unlearning only focuses on forgetting the entire class of data samples. For example, Lin et al. [23] added an entanglement-reduced structure into the model and then transferred the knowledge of the remaining data classes to the unlearned model. Chen et al. [3] shifted the decision boundary of the trained model to forget specific classes.

**Sample-wise machine unlearning.** The existing sample-wise MU works can be divided into two groups, *i.e., exact machine unlearning* and *approximate unlearning*. Bourtoule et al. [1] proposed SISA to unlearn a subset of the forgetting data samples by retraining the sub-models. Even though SISA improves efficiency compared to Retrain, plenty of sub-models still need to be retrained if the forgetting data samples are scattered. To improve the efficiency of exact MU methods, many state-of-the-art works [5, 9, 10, 22, 26, 27, 35] focus on approximate machine unlearning methods. For example, Golatkar et al. [9] and Mehta et al. [27] used the Fisher Information [26] to estimate the contributions. Foster et al. [8] used the Fisher information matrix to select and dampen the important parameter for the forgetting set. Moreover, Chundawat et al. [5] constructed a teacher-student model for MU. Graves et al. [10] stored the gradients related to the target unlearning data during the training process and then subtracted the gradients to update the model's parameters. Liu et al. [24] demonstrated that model sparsity can improve unlearning performance. Fan et al. [7] used the ascent gradients to construct a weight salience map to update only the specific weights rather than the entire model.

Different from the existing methods, we first analyze the specific conflict gradient pairs in the straightforward gradient ascent MU method and identify two key reasons: gradient direction and gradient domination. Furthermore, we propose the GDR method to mitigate the gradient direction problem. Specially, we propose a novel approximation method based on the gradient bank to simplify the large amount of gradient calculations. We also propose the GMA method to assign a dynamic weight parameter for gradients to handle the gradient dominant problem. Finally, we combine the GDR and GMA methods as GDR-GMA. By simultaneously handling both two problems, GDR-GMA achieves an effective, fidelity, and efficient MU method.

## 3 Preliminaries & Problem Analysis

In this section, we will first give a formulation of the sample-wise machine unlearning in Sec. 3.1 and describe the negative gradient method in Sec. 3.2. Then, we will analyze the gradient direction and gradient domination problems in Sec. 3.3 and Sec. 3.4, respectively.

### 3.1 Sample-wise Machine Unlearning

We first assume a sample space $X \subseteq \mathbb{R}^d$, the corresponding ground truth labels $\mathcal{Y} = \{1, 2, \ldots, C\}$ ($C$ is the number of data classes), and a training set $\mathcal{D} = \{(x, y)\}$, in which $x \subseteq X$ and $y \subseteq \mathcal{Y}$. We further define a forgetting dataset $\mathcal{D}_f \subseteq \mathcal{D}$, a remaining dataset $\mathcal{D}_r = \mathcal{D} \backslash \mathcal{D}_f$, and an original model with trainable parameters $\omega_0$, which trained on $\mathcal{D}$.

**Definition 1.** We define a machine learning algorithm, $\mathcal{A} : \mathcal{D} \to \omega_0$ and a machine unlearning method, $\mathcal{U} : \omega_0 \times \mathcal{D}_r \times \mathcal{D}_f \to \omega_u$. We denote the model that performed the unlearning operation as $\omega_u$ and the model trained with the remaining set $\mathcal{D}_r$ as $\omega_r$. The goal of MU is to attain an unlearned model $\omega_u$, wherein the knowledge of the forgetting data samples $\mathcal{D}_f$ equals that of the retrained model $\omega_r$. We can formulate it as:

$$\mathcal{K}(\mathcal{D}_f; \omega_u) = \mathcal{K}(\mathcal{D}_f; \omega_r), \tag{1}$$

in which $\mathcal{K}(\cdot)$ is the knowledge measuring function. Note that in the sample-wise machine unlearning task, the forgetting dataset can be a random subset of the training set, a subclass of data samples in a super-class, or a class of data samples.

### 3.2 Straightforward Gradient Ascent

Here, we will introduce the straightforward gradient ascent MU method, namely NegGrad. First, we perform forward propagation with a forgetting data sample $(x_f, y_f) \subseteq \mathcal{D}_f$, resulting in a prediction vector $\tilde{y}$. We calculate the cross entropy (CE) loss between the prediction vector $\tilde{y}_f$ and the ground truth label $y_f$. Then, we perform a backward propagation to calculate the gradient $g_f = \nabla_\omega \mathcal{L}_{\text{CE}}(\omega; x_f, y_f)$. Again, we can obtain the gradient $g_r$ of a remaining data sample $(x_r, y_r) \subseteq \mathcal{D}_r$. Finally, NegGrad updates the original model parameters as follows:

$$\omega_u \leftarrow \omega_0 + \eta(g_f - g_r), \tag{2}$$

in which $\eta$ is the learning rate. This process can be seen as multi-task learning. NegGrad has two main tasks: unlearning the forgetting dataset $\mathcal{D}_f$ and maintaining the remaining dataset $\mathcal{D}_r$. Specifically,

the ascending gradient $g_f$ is for the unlearning task, and the descent gradient $g_r$ is for the maintaining task.

### 3.3 Gradient Direction Problem

Here, we will analyze the gradient direction problem. First, we introduce an additional ascent gradient $g_F = \nabla_\omega \mathcal{L}_{\text{CE}}(\omega; x_F, y_F)$ of the other forgetting data samples $x_F \subseteq \mathcal{D}_f \backslash x_f$. Then, we define the gradient direction conflict as follows:

**Definition 2.** Given $\varphi_{ij}$ is the angle of two gradients $g_i$ and $g_j$. The two gradients have direction conflicts if $\cos \varphi_{ij} < 0$.

During the NegGrad unlearning process, we have three pairs of gradients as follows:

- $g_f$–$g_r$: the ascent gradient $g_f$ of the forgetting samples and the descent gradient $g_r$ of the remaining samples;
- $g_r$–$g_F$: the descent gradient $g_r$ of the remaining samples and the ascent gradient $g_F$ of the other forgetting samples.
- $g_f$–$g_F$: the ascent gradient $g_f$ of the forgetting samples and the ascent gradient $g_F$ of the other forgetting samples.

It should be noted that $g_F$ is not actually involved in the model update process. Moreover, there is a single pair $g_f$–$g_r$ and a set of pairs $\{g_r - g_{F_k}\}_{k=1}^{|\mathcal{D}_F|}$ and $\{g_f - g_{F_k}\}_{k=1}^{|\mathcal{D}_F|}$ in an update step because we need to consider the gradient of each sample in $\mathcal{D}_F$.

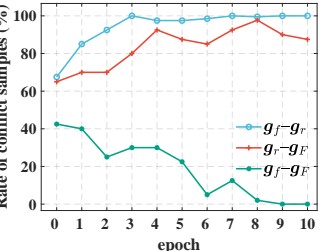

**Figure 1: Measurement of the percentage conflicting samples on CIFAR-10 using ResNet-18. The percentage maintains high for $g_f$–$g_r$ (blue) and $g_r$–$g_F$ (red). The percentage tends to be low for $g_f$–$g_F$ (green).**

Based on Definition 2, we empirically measure the percentage of data samples having these three direction conflicts during the update of the NegGrad method. As shown in Fig. 1, these observations indicate low conflict for the $g_f$–$g_F$ pair but evident conflict for both $g_r$–$g_F$ and $g_f$–$g_F$ pairs.

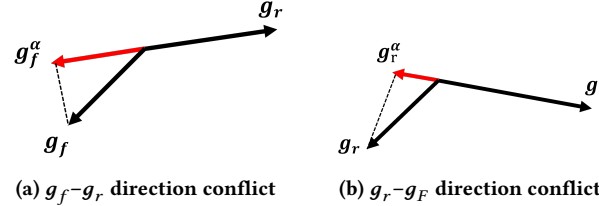

(a) $g_f$–$g_r$ direction conflict      (b) $g_r$–$g_F$ direction conflict

**Figure 2: Illustrations of the gradient direction problems. $g^\alpha$ denotes the orthogonal gradient component of $g$.**

When these direction conflicts occur, we will explain the negative impact on performance with two specific illustrations.

**Direction conflict of $g_f$–$g_r$.** As shown in Fig. 2(a), direction of the gradient component $g_f^\alpha$ is opposite to $g_r$. As a result, updating the model with $g_f$ will maximize $\mathcal{L}_{CE}(\omega; x_r, y_r)$ so that the model tends to unlearn the remaining sample $(x_r, y_r)$. Similarly, updating the model with $g_r$ tends to maintain the forgetting data sample $(x_f, y_f)$. These direction conflicts will also cause conflicts between the unlearning task and the maintaining task to decline the performance.

**Direction conflict of $g_r$–$g_F$.** As shown in Fig. 2(b), $g_r$ and $g_F$ have direction conflict so that updating the model with $g_r$ will also minimize $\mathcal{L}_{CE}(\omega; x_F, y_F)$, *i.e.*, the model tends to maintain the forgetting data sample $(x_F, y_F)$ rather than unlearn it.

### 3.4 Gradient Domination Problem

The MU process can be seen as multi-task learning, and there are two tasks in the MU process: the forgetting task and the maintaining task. The forgetting task aims to forget the forgetting set, while the maintaining task aims to maintain the remaining set. Ideally, the related gradients of each task jointly update the model to learn the specific knowledge of each task. However, the gradient domination problem is a common challenge that the model's updates during training are biased towards optimizing for one task over others, *i.e.*, a single gradient dominates the model update.

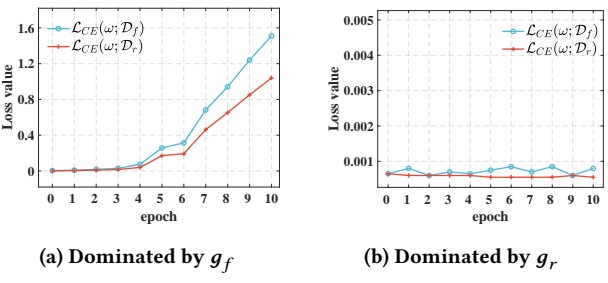

(a) Dominated by $g_f$      (b) Dominated by $g_r$

**Figure 3: Measurement of the loss values on CIFAR-10 using ResNet-18 when the gradient dominant problem occurs. Both of the forgetting loss $\mathcal{L}_{CE}(\omega; \mathcal{D}_f)$ and the remaining loss $\mathcal{L}_{CE}(\omega; \mathcal{D}_r)$ increase when dominated by $g_f$. Both loss values are maintained almost unchanged when dominated by $g_r$.**

We empirically evaluate this problem through the values of the loss function. As shown in Fig. 3 (a), increasing the remaining loss will decrease the model classification performance. In contrast, the barely changed forgetting loss will lead to invalid forgetting, as shown in Fig. 3 (b). Inspired by the previous work, Yu et al. [36] pointed out that the magnitude of gradients will affect the gradient domination in MTL. Therefore, we try to evaluate the effect of magnitude on this dominant problem by assigning a constant magnitude parameter to the gradients. As shown in Table 1, the results show that the gradient dominant problem always occurs during the NegGrad unlearning process, resulting in unchanged or increased loss values.

### 4 Proposed Method

In Sec. 4.1, we propose the Gradient Direction Rectification (GDR) method for the gradient direction problem. In Sec. 4.2, we propose

**Table 1: Effect of the magnitude on the gradient dominant problem by assigning different weight parameters. " – " denotes that the loss values are almost unchanged, and "↑ " denotes that both the loss values are increased.**

| Magnitude Weight | | Dominant | Loss values |
|---|---|---|---|
| $g_f$ | $g_r$ | | |
| 0.1 | 0.9 | $g_r$ | – |
| 0.2 | 0.8 | $g_r$ | – |
| 0.3 | 0.7 | $g_r$ | – |
| 0.4 | 0.6 | $g_f$ | ↑ |
| 0.5 | 0.5 | $g_f$ | ↑ |
| 0.6 | 0.4 | $g_f$ | ↑ |
| 0.7 | 0.3 | $g_f$ | ↑ |
| 0.8 | 0.2 | $g_f$ | ↑ |
| 0.9 | 0.1 | $g_f$ | ↑ |

the Gradient Magnitude Adjustment (GMA) method for the domination problem. Finally, we will describe the combined GDR-GMA for the sample-wise MU task in Sec. 4.3.

### 4.1 Gradient Direction Rectification (GDR)

In Sec. 3.3, we define three gradients that affect the NegGrad MU process, *i.e.*, $g_f$, $g_r$, and $g_F$. However, it is very time-consuming to calculate the set of gradients $\{g_{F_k}\}_{k=1}^{|\mathcal{D}_F|}$. To address this problem, we empirically measure the cosine similarity between gradients of the same sample in adjacent epochs.

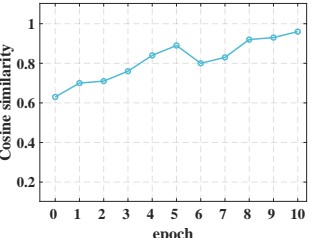

**Figure 4: The cosine similarity between gradients in adjacent epochs remains high during the MU process on CIFAR-10 using ResNet-18.**

As shown in Fig. 4, this observation indicates that the gradients of the same sample in the adjacent epoch are highly similar during the MU process. Based on this observation, we use a gradient bank to replace the repeated calculation process with storing the gradients in the previous epoch. We formulate this replacement process as:

$$\{g_{B_k}^e\}_{k=1}^{|Bank|} := \{g_{F_k}^{e-1}\}_{k=1}^{|\mathcal{D}_F|}, \qquad (3)$$

where $e$ denotes the current epoch, $e-1$ denotes the previous epoch, and $|Bank|$ is the size of gradient bank, which equals to $|\mathcal{D}_F|$.

We propose the Gradient Direction Rectification (GDR) method to alleviate these direction conflicts by rectifying the direction between the two conflicting gradients $g_i$ and $g_j$ if $\cos \varphi_{ij} < 0$. After rectification of the direction, the gradient $i$ is projected onto the orthonormal plane of the conflicting gradient $j$ to remove destructive conflicts, as shown in Fig. 6. Formally, GDR projects the gradient $i$

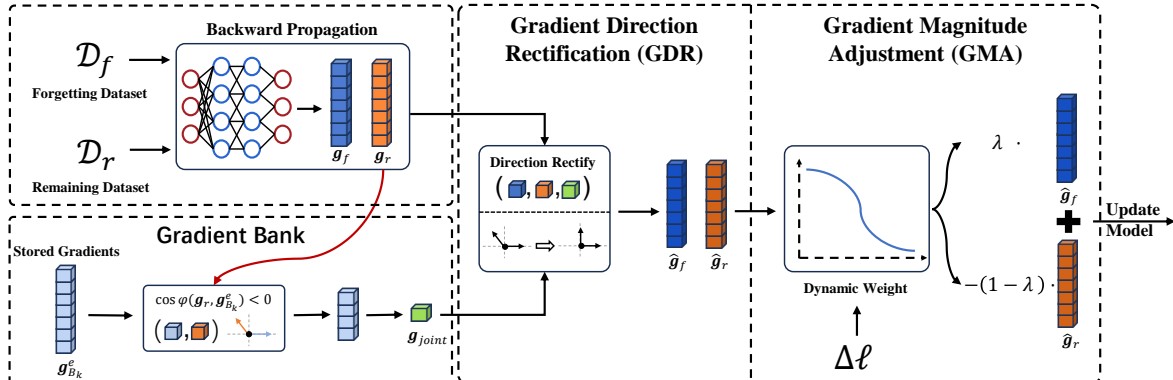

**Figure 5: An overview of our proposed GDR-GMA unlearning method. First, we calculate gradients $g_f$ and $g_r$ and get the joint gradient $g_{joint}$ from the gradient bank. Then, GDR rectifies the gradient directions of the conflicting gradients and get the rectified gradients $\hat{g}_f$ and $\hat{g}_r$. GMA calculates the dynamic weight parameter $\lambda$ using the loss deviation $\Delta\ell$ to adjust the magnitude of the update gradients. Finally, we update the model using the combined gradients $\hat{g}_f - (1 - \lambda_t)\hat{g}_r$.**

onto the orthonormal plane of the gradient $j$ as:

$$\hat{g}_i := g_i - \frac{g_i \cdot g_j}{||g_j||^2} g_j, \qquad (4)$$

where $||g||$ denotes the L2-norm of the gradient $g$.

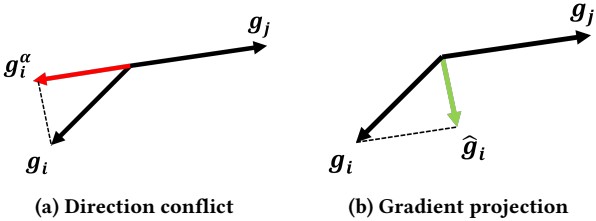

| (a) Direction conflict | (b) Gradient projection |
|---|---|

**Figure 6: When the direction conflict occurs (a), the GDR method will project the gradient onto the orthonormal plane of the conflicting gradient (b).**

As described in Sec. 3.3, we analyze the direction conflict problem only exists between the pairs of $g_f$–$g_r$ and $g_f$–$g_F$. However, multiple projections will be needed to alleviate direction conflicts. To avoid the excessive direction deviation caused by the multiple projections, we define the joint conflict gradients in the gradient bank as:

$$g_{joint} = \frac{\sum_{k=1}^{|Bank|} \mathbf{1}(\cos \varphi(g_r, g_{B_k}^e) < 0) \cdot g_{B_k}^e}{\sum_{k=1}^{|Bank|} \mathbf{1}(\cos \varphi(g_r, g_{B_k}^e) < 0)}, \qquad (5)$$

where $\mathbf{1}(g_i < 0)$ is an indicator function which yields a value of 1 if $g_i < 0$ and 0 otherwise. Then, GDR can once project $g_r$ onto the plane of the joint gradient instead of the multiple projections.

We present the GDR method in Algorithm 1. The proposed algorithm utilize three types of gradients $g_f$, $g_r$, and $g_F$. We use an approximate joint gradient $g_{joint}$ from the gradient bank to reduce computational overhead and avoid multiple projections. If conflicts are detected, we project the gradient based on Eq. 4 to alleviate direction conflicts.

---

**Algorithm 1:** Gradient Direction Rectification (GDR)

---

**1** **for** *e in epochs* **do**

**2**     $g_f \leftarrow \nabla_{\omega_t} \mathcal{L}_{CE}(\omega_t; x_f, y_f)$

**3**     $g_r \leftarrow \nabla_{\omega_t} \mathcal{L}_{CE}(\omega_t; x_r, y_r)$

**4**     $g_{joint} \leftarrow \dfrac{\sum_{k=1}^{|Bank|} \mathbf{1}(\cos \varphi(g_r, g_{B_k}^e) < 0) \cdot g_{B_k}^t}{\sum_{k=1}^{|Bank|} \mathbf{1}(\cos \varphi(g_r, g_{B_k}^e) < 0)}$

**5**     **if** $\cos \varphi(g_f, g_r) < 0$ **then**

**6**       $\hat{g}_f \leftarrow g_f - \frac{g_f \cdot g_r}{||g_r||^2} g_r$

**7**       $\hat{g}_r \leftarrow g_r - \frac{g_r \cdot g_f}{||g_f||^2} g_f$

**8**     **end**

**9**     **if** $\cos \varphi(\hat{g}_r, g_{joint}) < 0$ **then**

**10**       $\hat{g}_r \leftarrow \hat{g}_r - \frac{\hat{g}_r \cdot g_{joint}}{||g_{joint}||^2} g_{joint}$

**11**     **end**

**12** **end**

---

## 4.2 Gradient Magnitude Adjustment (GMA)

In Sec. 3.4, we empirically observe that the magnitude will affect the gradient domination. However, simply adjusting the magnitude cannot balance the two gradients and instead results in domination by a single gradient, as shown in Table 1. Therefore, we propose a Gradient Magnitude Adjustment (GMA) method to dynamically adjust the magnitude. Specifically, GMA can dynamically adjust the magnitude of update gradients according to the three states of the MU process as follows:

- **State I (Maintaining Dominance):** In this state, the remaining loss value maintains nearly the same as that in the original model, which shows that the remaining gradient $g_r$ dominates the MU process. Hence, GMA will assign a large magnitude to the forgetting gradient $g_f$ to balance the forgetting and maintaining tasks.
- **State II (Balance):** In this state, none of the dominance occurs. Hence, GMA needs to flexibly adjust the magnitude of the update gradients to maintain this balance.

- **State III (Forgetting Dominance):** When the remaining loss dramatically deviates from the original loss value, GMA will assign a large magnitude to the remaining gradient $g_r$ to force the model to maintain the remaining samples.

Formally, we define the dynamic magnitude weight (DMW) as:

$$\lambda_t = \frac{1}{1 + \exp(\gamma \cdot (\Delta\ell_t - \epsilon))}, \tag{6}$$

where $t$ denotes the $t$-th unlearning step, $\epsilon$ is a constant small value, $\gamma$ denotes the steepness parameter, and the loss deviation $\Delta\ell_t = |\mathcal{L}_{CE}(\omega_u^t; x_r^t) - \mathcal{L}_{CE}(\omega_0; \mathcal{D}_r)|$ denotes the difference between the loss values of the remaining samples in the $t$-th step and the remaining dataset in the original model. The steepness parameter $\gamma$ controls how sharply the weight parameter transitions from its minimum to maximum value. A large $\epsilon$ will tolerate a larger increase in the loss values of the remaining samples. With the DMW parameter $\lambda_t$, GMA assigns it to gradients in the $t$-th update step as:

$$\omega_u^{t+1} \leftarrow \omega_u^t + \eta[\lambda_t g_f^t - (1 - \lambda_t)g_r^t]. \tag{7}$$

Furthermore, we will describe why the above proposed DMW parameter can suit these three states. As shown in Fig. 7, we illustrate the relationship between the DMW parameter $\lambda_t$ and the loss deviation $\Delta\ell_t$ in the three states. In State I, GMA adjusts $\lambda_t$ to be close to 1 so that the forgetting loss will rapidly ascend to the balance state. A low decay rate is used in this state to make $\lambda_t$ insensitive to the loss deviation and close to 1. In State II, when the loss deviation $\Delta\ell_t$ is around the small value $\epsilon$, GMA flexibly adjusts the magnitude of $g_f$ and $g_r$ with a high change rate of $\lambda_t$ to keep the balance between forgetting and maintaining tasks. In State III, when the loss deviation $\Delta\ell_t$ is greater than the small value $\epsilon$, GMA adjusts the magnitude of $g_r$ to close to 1 with a low change rate of $\lambda_t$. With a large magnitude of $g_r$, GMA can make the model tend to maintain the remaining samples and get back to the balance state as soon as possible.

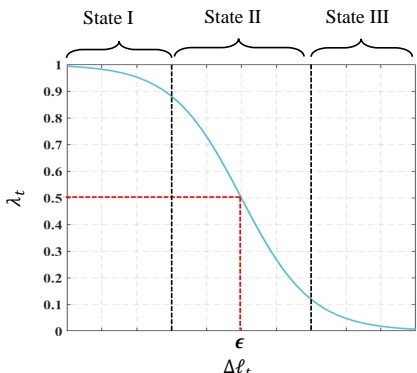

**Figure 7: Illustration of the relationship between the dynamic weight parameter $\lambda_t$ and the loss deviation $\Delta\ell_t$ in the three states. State I: $\Delta\ell_t$ is close to 0 and $\lambda_t$ tends to be 1 with a low decay rate; State II: $\Delta\ell_t$ is around the preset small value $\epsilon$ and $\lambda_t$ is sensitive to $\Delta\ell_t$ a high decay rate; State III: $\Delta\ell_t$ is much higher than $\epsilon$ and $\lambda_t$ will be close to 0 with a low decay rate.**

Implementing GMA is simple and barely brings additional computational overhead because it only requires numerical loss values. Through the crafted DMW parameter, GMA can dynamically adjust the update gradients' magnitude to balance these two tasks.

---

**Algorithm 2:** GDR-GMA Unlearning Process

---

1 **for** $e$ in epochs **do**
2     **for** $t$ in steps **do**
        // Get the rectified gradients by GDR
3         $\hat{g}_f, \hat{g}_r \leftarrow$ GDR
        // Get the dynamic weight by GMA
4         $\lambda_t \leftarrow$ GMA
        // Update the model
5         $\omega_u^{t+1} \leftarrow \omega_u^t + \eta[\lambda_t \hat{g}_f^t - (1 - \lambda_t)\hat{g}_r^t].$
6     **end**
7 **end**

---

## 4.3 Machine Unlearning via GDR-GMA

In Sec. 4.1, we propose the GDR method to rectify the direction of two pairs of conflicting gradients. Despite alleviating the direction conflicts, GDR cannot address the gradient dominant problem. Therefore, in Sec. 4.2, we propose the GMA method to dynamically adjust the magnitude of the update gradients to balance the forgetting task and the remaining task.

We present the combined GDR-GMA method in Algorithm 2. First, GDR-GMA rectifies the direction of the conflicting gradients by the GDR method. Then, GDR-GMA calculates the dynamic weight parameter using the GMA method. Finally, GDR-GMA updates the model with the rectified gradients and dynamic magnitude weight parameters.

## 5 Experiments

In this Section, we conduct extensive experiments to empirically evaluate the proposed GDR-GMA method. We compare its performance with several baseline MU methods in three MU scenarios: **random data forgetting**, **subclass forgetting**, and **class forgetting**. Furthermore, we also conduct ablation experiments to prove the effectiveness of GDR-GMA.

## 5.1 Experiment Setups

**Datasets and models.** In this paper, we evaluate GDR-GMA on three datasets, CIFAR-10, CIFAR-20 [20], and Tiny-ImageNet [21], using two model architectures, ResNet-18 [13] and Vision Transformer (ViT) [6].

**Baseline methods.** In random data forgetting and subclass forgetting, we compare GDR-GMA with the following baselines: **Retrain**, **Fine-tune**, **Random Labels** [9], **NegGrad**, **Fisher** [9], **Unrolling** [33], **BadT** [5], $L_1$-**Sparse** [24], **SalUn** [7], **SSD** [32]. Furthermore, we add two new baselines in class forgetting, *i.e.,* **ERM-KTP** [23] and **Boundary** [3].

**Metrics.** Following previous works [5, 7, 32], we use $Acc_{\mathcal{D}_f}$, $Acc_{\mathcal{D}_r}$, and $Acc_{val}$ to represent the classification accuracy on the forgetting set, the remaining set, and the validation set, respectively. Besides,

**Table 2: Performance comparison with several baselines in 20% random data forgetting and 50% random data forgetting. The performance gap against Retrain is provided in ($\Delta$). The optimal Avg. Gap is marked in red.**

| Dataset & Model | Approach | Random Data Forgetting (20%) | | | | | | Random Data Forgetting (50%) | | | | | |
|---|---|---|---|---|---|---|---|---|---|---|---|---|---|
| | | $Acc_{\mathcal{D}_f}(\Delta \downarrow)$ | $Acc_{\mathcal{D}_r}(\Delta \downarrow)$ | $Acc_{val}(\Delta \downarrow)$ | MIA $(\Delta \downarrow)$ | Avg. Gap $\downarrow$ | RTE $\downarrow$ | $Acc_{\mathcal{D}_f}(\Delta \downarrow)$ | $Acc_{\mathcal{D}_r}(\Delta \downarrow)$ | $Acc_{val}(\Delta \downarrow)$ | MIA $(\Delta \downarrow)$ | Avg. Gap $\downarrow$ | RTE $\downarrow$ |
| CIFAR-10 ResNet-18 | Retrain | $94.00_{\pm0.54}(0.00)$ | $100.00_{\pm0.00}(0.00)$ | $94.00_{\pm0.12}(0.00)$ | $76.76_{\pm0.04}(0.00)$ | 0.00 | 28.53 | $91.73_{\pm0.15}(0.00)$ | $100.00_{\pm0.00}(0.00)$ | $91.99_{\pm0.12}(0.00)$ | $70.79_{\pm0.03}(0.00)$ | 0.00 | 19.71 |
| | Fine-tune | $85.82_{\pm1.12}(8.18)$ | $88.07_{\pm0.24}(11.93)$ | $86.98_{\pm0.87}(7.02)$ | $32.09_{\pm0.14}(44.67)$ | 17.95 | 1.48 | $87.48_{\pm1.52}(4.25)$ | $88.98_{\pm0.44}(11.02)$ | $86.78_{\pm0.26}(5.21)$ | $66.81_{\pm0.31}(3.98)$ | 6.12 | 1.04 |
| | Random Labels | $81.58_{\pm1.12}(12.42)$ | $85.67_{\pm1.06}(14.33)$ | $83.60_{\pm0.21}(10.40)$ | $59.69_{\pm0.25}(17.07)$ | 13.56 | 1.21 | $81.45_{\pm0.66}(10.28)$ | $84.51_{\pm0.56}(15.49)$ | $82.44_{\pm0.12}(9.55)$ | $45.35_{\pm0.29}(25.44)$ | 15.19 | 2.19 |
| | NegGrad | $72.81_{\pm0.89}(21.19)$ | $76.58_{\pm1.61}(23.42)$ | $71.48_{\pm1.64}(22.52)$ | $76.25_{\pm0.89}(0.51)$ | 16.91 | 1.33 | $72.69_{\pm1.57}(19.04)$ | $75.70_{\pm1.49}(24.30)$ | $70.50_{\pm1.12}(21.49)$ | $76.54_{\pm1.23}(5.75)$ | 17.64 | 2.07 |
| | Fisher [9] | $23.28_{\pm2.56}(70.72)$ | $23.96_{\pm3.41}(76.04)$ | $23.10_{\pm3.27}(70.90)$ | $62.23_{\pm0.28}(14.53)$ | 58.05 | 117.82 | $22.39_{\pm1.45}(69.34)$ | $22.63_{\pm2.15}(77.37)$ | $22.47_{\pm2.31}(69.52)$ | $62.64_{\pm0.53}(8.15)$ | 61.10 | 76.01 |
| | Unrolling [33] | $92.97_{\pm0.55}(1.03)$ | $93.72_{\pm0.38}(6.28)$ | $87.58_{\pm0.11}(6.42)$ | $44.30_{\pm0.24}(32.46)$ | 11.55 | 0.27 | $89.73_{\pm0.23}(10.27)$ | $85.34_{\pm0.21}(6.65)$ | $85.34_{\pm0.21}(6.65)$ | $43.66_{\pm0.14}(27.13)$ | 12.66 | 0.44 |
| | BadT [5] | $87.12_{\pm0.15}(6.88)$ | $91.74_{\pm0.39}(8.26)$ | $87.83_{\pm0.16}(6.17)$ | $47.81_{\pm0.14}(28.95)$ | 12.57 | 1.12 | $85.93_{\pm0.28}(5.80)$ | $89.48_{\pm0.32}(10.52)$ | $85.78_{\pm0.20}(6.21)$ | $27.35_{\pm0.29}(43.44)$ | 16.50 | 1.14 |
| | SSD [8] | $97.72_{\pm0.72}(3.72)$ | $97.51_{\pm0.04}(2.49)$ | $91.73_{\pm0.46}(2.27)$ | $88.04_{\pm0.46}(11.28)$ | 4.94 | 3.04 | $99.99_{\pm0.01}(8.26)$ | $100.00_{\pm0.00}(0.00)$ | $94.98_{\pm0.25}(2.99)$ | $92.04_{\pm0.88}(21.25)$ | 8.12 | 3.17 |
| | $L_1$-Sparse [24] | $92.72_{\pm0.62}(1.28)$ | $96.81_{\pm0.30}(3.19)$ | $91.62_{\pm0.58}(2.38)$ | $71.28_{\pm0.04}(5.48)$ | 3.08 | 2.25 | $90.81_{\pm0.81}(0.92)$ | $94.11_{\pm0.26}(5.89)$ | $88.37_{\pm0.73}(3.62)$ | $65.75_{\pm0.22}(5.04)$ | 3.87 | 1.39 |
| | SalUn [7] | $93.87_{\pm0.66}(0.13)$ | $99.17_{\pm0.21}(0.83)$ | $92.53_{\pm0.54}(1.47)$ | $68.47_{\pm0.98}(8.29)$ | 2.68 | 2.77 | $92.46_{\pm0.59}(0.73)$ | $98.06_{\pm0.51}(1.94)$ | $89.93_{\pm0.06}(2.06)$ | $62.88_{\pm0.81}(7.91)$ | 3.16 | 1.67 |
| | GDR-GMA (ours) | $93.84_{\pm0.04}(0.16)$ | $99.22_{\pm0.11}(0.78)$ | $92.40_{\pm1.47}(1.60)$ | $82.36_{\pm0.06}(5.60)$ | **2.04** | 1.40 | $92.10_{\pm0.08}(0.63)$ | $99.07_{\pm0.45}(0.93)$ | $90.37_{\pm1.10}(1.62)$ | $76.27_{\pm0.18}(5.48)$ | **2.17** | 2.12 |
| Tiny-ImageNet ViT | Retrain | $52.87_{\pm0.62}(0.00)$ | $90.29_{\pm0.71}(0.00)$ | $54.18_{\pm0.70}(0.00)$ | $47.27_{\pm0.08}(0.00)$ | 0.00 | 846.71 | $47.75_{\pm0.43}(0.00)$ | $92.46_{\pm0.52}(0.00)$ | $48.54_{\pm0.37}(0.00)$ | $44.83_{\pm0.22}(0.00)$ | 0.00 | 602.18 |
| | Fine-tune | $70.85_{\pm0.93}(17.98)$ | $88.30_{\pm0.85}(1.99)$ | $52.71_{\pm0.99}(1.47)$ | $53.81_{\pm0.39}(6.54)$ | 7.00 | 27.04 | $70.75_{\pm0.98}(23.00)$ | $92.33_{\pm0.50}(0.13)$ | $51.76_{\pm0.91}(3.22)$ | $51.70_{\pm0.49}(6.87)$ | 8.30 | 17.74 |
| | Random Labels | $75.76_{\pm0.76}(22.89)$ | $82.71_{\pm0.31}(7.58)$ | $52.83_{\pm0.75}(1.35)$ | $27.87_{\pm0.28}(19.40)$ | 12.81 | 16.86 | $79.99_{\pm0.88}(32.24)$ | $82.33_{\pm0.53}(10.13)$ | $52.83_{\pm0.76}(4.29)$ | $34.35_{\pm0.65}(10.48)$ | 14.28 | 41.66 |
| | NegGrad | $84.58_{\pm1.19}(31.71)$ | $85.56_{\pm0.58}(4.73)$ | $51.52_{\pm1.26}(2.66)$ | $63.91_{\pm0.34}(16.64)$ | 13.93 | 13.17 | $51.19_{\pm0.66}(3.44)$ | $51.79_{\pm1.47}(40.67)$ | $36.34_{\pm1.28}(12.20)$ | $53.62_{\pm1.21}(8.79)$ | 16.27 | 23.57 |
| | Unrolling [33] | $92.66_{\pm0.68}(39.79)$ | $92.79_{\pm0.39}(2.50)$ | $51.52_{\pm1.26}(2.66)$ | $68.37_{\pm0.54}(21.10)$ | 15.92 | 7.83 | $89.06_{\pm0.64}(41.31)$ | $54.35_{\pm0.27}(38.11)$ | $41.91_{\pm0.56}(6.63)$ | $26.30_{\pm0.34}(18.53)$ | 17.47 | 19.52 |
| | BadT [5] | $48.20_{\pm0.49}(4.67)$ | $57.73_{\pm0.30}(32.56)$ | $43.24_{\pm0.47}(10.94)$ | $29.45_{\pm0.25}(17.82)$ | 16.50 | 10.32 | $45.78_{\pm0.51}(1.97)$ | $54.35_{\pm0.27}(38.11)$ | $41.91_{\pm0.56}(6.63)$ | $26.30_{\pm0.34}(18.53)$ | 16.31 | 10.43 |
| | SSD [8] | $92.97_{\pm0.77}(40.10)$ | $92.75_{\pm0.24}(2.46)$ | $54.35_{\pm0.68}(0.17)$ | $68.50_{\pm0.26}(21.23)$ | 15.99 | 25.64 | $92.80_{\pm0.45}(45.05)$ | $92.83_{\pm0.28}(0.37)$ | $54.35_{\pm0.79}(5.81)$ | $68.25_{\pm0.21}(23.42)$ | 18.66 | 27.70 |
| | $L_1$-Sparse [24] | $68.71_{\pm0.57}(15.84)$ | $85.13_{\pm0.21}(5.16)$ | $50.25_{\pm0.50}(3.93)$ | $44.26_{\pm0.32}(3.01)$ | 6.99 | 32.15 | $69.04_{\pm1.47}(21.29)$ | $87.55_{\pm0.78}(4.91)$ | $49.78_{\pm1.06}(1.24)$ | $43.32_{\pm0.70}(1.51)$ | 7.24 | 20.17 |
| | SalUn [7] | $57.62_{\pm0.94}(4.75)$ | $91.46_{\pm0.89}(1.17)$ | $53.42_{\pm0.12}(0.76)$ | $43.36_{\pm0.04}(3.91)$ | 2.65 | 34.83 | $53.79_{\pm0.96}(6.04)$ | $94.12_{\pm0.64}(1.66)$ | $52.98_{\pm0.19}(4.44)$ | $42.14_{\pm0.09}(2.69)$ | 3.71 | 24.38 |
| | GDR-GMA (ours) | $51.69_{\pm0.48}(1.18)$ | $91.49_{\pm0.02}(1.20)$ | $51.78_{\pm0.06}(2.40)$ | $52.91_{\pm0.16}(5.64)$ | **2.61** | 13.67 | $46.25_{\pm0.28}(1.50)$ | $90.45_{\pm0.23}(2.01)$ | $45.06_{\pm0.04}(3.48)$ | $50.92_{\pm0.22}(6.09)$ | **3.27** | 25.60 |

we leverage the membership inference attack (MIA) on the forgetting set to measure whether the forgetting data samples are in the training set. We use RTE to measure the time overhead in minutes of the unlearning process. We use $Acc_{\mathcal{D}_f}$ and MIA to measure the effectiveness of MU methods, $Acc_{\mathcal{D}_r}$ and $Acc_{val}$ to evaluate fidelity, and RTE to measure efficiency. To present a summary performance gap against the ideal baseline Retrain, we introduce the Avg. Gap by calculating the average performance gaps in $Acc_{\mathcal{D}_f}$, $Acc_{\mathcal{D}_r}$, $Acc_{val}$ and MIA. Note that **the better performance** of an MU method corresponds to **the smaller performance gap** with **Retrain**. The results are given by a format $a_{\pm b}$ with mean $a$ and standard deviation $b$ over ten independent experiments.

**Hyper-parameters.** The original models are trained for 200 epochs using the SGD optimizer with a momentum of 0.9, weight decay of 5e-4, and an initial learning rate of 0.1, divided by 10 after 100 and 150 epochs, respectively. For our proposed GDR-GMA, we set the steepness parameter $\gamma$ to 100 and the constant small value $\epsilon$ to 0.02.

## 5.2 Evaluation of GDR-GMA

**Comparison experiments in random data forgetting.** We conduct extensive comparison experiments with several baselines. First, we evaluate the performance of forgetting a random subset of data samples. Following previous works [5, 7, 32], we consider two unlearning scenarios, *i.e.,* 20% random data forgetting and 50% random data forgetting. Based on the results presented in Table 2, we draw the following three key observations:

First, following the previous work [7], Avg. Gap is a more comprehensive metric to evaluate the performance of the MU methods. Some methods may be the strongest when considering only a single metric, but this comes at the cost of sacrificing the other metrics. However, GDR-GMA still achieves the smallest average performance gap against Retrain on these two (data-model) setups in both scenarios, demonstrating its superior effectiveness and fidelity.

Second, GDR-GMA inherits the efficiency of the gradient ascent methods while maintaining the model classification performance. GDR-GMA significantly improves the performance of NegGrad with negligible additional computation overhead. Furthermore, GDR-GMA has a competitive computation efficiency with these baselines, as evidenced by the RTE metric.

Third, randomly forgetting 50% data samples is a more complex scenario, resulting in a higher average performance gap than that in 20% random data forgetting scenario. Besides, the MU methods on Tiny-ImageNet using ViT are more demanding due to the larger scale of data samples and model parameters. Nevertheless, GDR-GMA still has the smallest average performance gap and achieves superior performance on both effectiveness and fidelity.

**Table 3: Performance comparison with several baselines on CIFAR-20 using ResNet-18 in subclass forgetting.**

| Approach | $Acc_{\mathcal{D}_f}(\Delta \downarrow)$ | $Acc_{\mathcal{D}_r}(\Delta \downarrow)$ | $Acc_{val}(\Delta \downarrow)$ | MIA $(\Delta \downarrow)$ | Avg. Gap $\downarrow$ | RTE $\downarrow$ |
|---|---|---|---|---|---|---|
| Retrain | $2.52_{\pm0.62}(0.00)$ | $100.00_{\pm0.00}(0.00)$ | $68.94_{\pm0.62}(0.00)$ | $0.00_{\pm0.00}(0.00)$ | 0.00 | 35.74 |
| Fine-tune | $3.33_{\pm0.14}(0.81)$ | $90.64_{\pm0.21}(9.36)$ | $68.22_{\pm0.19}(0.72)$ | $0.00_{\pm0.00}(0.00)$ | 2.72 | 1.79 |
| Random Labels | $4.80_{\pm0.17}(2.28)$ | $69.97_{\pm0.38}(30.03)$ | $62.13_{\pm0.23}(6.81)$ | $0.00_{\pm0.00}(0.00)$ | 9.78 | 1.13 |
| NegGrad | $31.30_{\pm0.15}(28.78)$ | $97.57_{\pm0.37}(2.43)$ | $80.04_{\pm0.02}(11.10)$ | $0.09_{\pm0.01}(0.09)$ | 10.60 | 0.52 |
| Unrolling [33] | $15.07_{\pm0.04}(12.55)$ | $97.30_{\pm0.13}(2.70)$ | $83.15_{\pm0.52}(14.21)$ | $0.31_{\pm0.02}(0.31)$ | 7.44 | 0.43 |
| BadT [5] | $7.77_{\pm0.20}(5.25)$ | $94.81_{\pm0.74}(5.19)$ | $72.15_{\pm0.07}(3.21)$ | $0.10_{\pm0.01}(0.10)$ | 3.44 | 1.85 |
| SSD [8] | $0.10_{\pm0.00}(2.42)$ | $70.45_{\pm0.48}(29.55)$ | $59.78_{\pm0.18}(9.16)$ | $32.31_{\pm0.28}(32.31)$ | 18.36 | 3.25 |
| $L_1$-Sparse [24] | $3.70_{\pm0.11}(1.18)$ | $86.58_{\pm0.25}(13.42)$ | $67.99_{\pm0.23}(0.95)$ | $0.00_{\pm0.00}(0.00)$ | 3.89 | 1.94 |
| SalUn [7] | $2.68_{\pm0.96}(0.16)$ | $96.21_{\pm0.60}(3.79)$ | $66.89_{\pm0.06}(2.05)$ | $0.00_{\pm0.00}(0.00)$ | 1.50 | 2.16 |
| GDR-GMA (ours) | $2.01_{\pm0.09}(0.51)$ | $99.76_{\pm0.02}(0.24)$ | $67.99_{\pm0.36}(0.95)$ | $0.00_{\pm0.00}(0.00)$ | **0.42** | 0.60 |

**Comparison experiments in subclass forgetting.** We then explore the performance of forgetting a subclass of data samples, following previous works [5, 8]. Sub-class forgetting is a simpler MU scenario than random data forgetting because sub-class samples are more similar. This simple data distribution makes the model more easily unlearn the forgetting data samples while maintaining the remaining data samples. As shown in Table 3, GDR-GMA achieves a superior performance in sub-class forgetting than random data forgetting. Moreover, GDR-GMA has the smallest performance gap and significantly outperforms the state-of-the-art works regarding time overhead.

**Table 4: Performance comparison with several baselines on CIFAR-10 using ResNet-18 in class forgetting.**

| Approach | $Acc_{\mathcal{D}_f}(\Delta \downarrow)$ | $Acc_{\mathcal{D}_r}(\Delta \downarrow)$ | $Acc_{val}(\Delta \downarrow)$ | MIA $(\Delta \downarrow)$ | Avg. Gap $\downarrow$ | RTE $\downarrow$ |
|---|---|---|---|---|---|---|
| Retrain | $0.00_{\pm0.00}(0.00)$ | $100.00_{\pm0.00}(0.00)$ | $79.64_{\pm0.62}(0.00)$ | $7.30_{\pm0.00}(0.00)$ | 0.00 | 33.20 |
| Fine-tune | $0.00_{\pm0.00}(0.00)$ | $96.47_{\pm0.54}(3.53)$ | $78.43_{\pm0.61}(1.21)$ | $24.22_{\pm0.45}(16.92)$ | 5.41 | 1.81 |
| Random Labels | $0.00_{\pm0.00}(0.00)$ | $86.97_{\pm0.78}(13.03)$ | $84.70_{\pm0.53}(5.06)$ | $0.28_{\pm0.04}(7.02)$ | 6.28 | 0.87 |
| NegGrad | $19.36_{\pm0.60}(19.36)$ | $97.66_{\pm0.96}(2.34)$ | $84.29_{\pm0.04}(4.65)$ | $22.14_{\pm0.98}(14.84)$ | 10.30 | 0.71 |
| Unrolling [33] | $1.56_{\pm0.21}(1.56)$ | $84.10_{\pm0.86}(15.90)$ | $78.69_{\pm0.43}(0.95)$ | $16.28_{\pm0.09}(8.98)$ | 6.85 | 0.31 |
| BadT [5] | $0.00_{\pm0.00}(0.00)$ | $94.94_{\pm0.54}(5.06)$ | $80.79_{\pm0.91}(1.15)$ | $0.00_{\pm0.00}(7.30)$ | 3.38 | 1.41 |
| SSD [8] | $0.00_{\pm0.00}(0.00)$ | $94.65_{\pm0.14}(5.35)$ | $92.55_{\pm0.80}(12.91)$ | $0.00_{\pm0.00}(7.30)$ | 6.39 | 3.10 |
| $L_1$-Sparse [24] | $0.00_{\pm0.00}(0.00)$ | $90.07_{\pm0.14}(9.93)$ | $79.70_{\pm0.88}(0.06)$ | $13.92_{\pm0.23}(6.62)$ | 4.15 | 1.65 |
| SalUn [7] | $0.00_{\pm0.00}(0.00)$ | $95.89_{\pm0.19}(4.11)$ | $82.76_{\pm0.28}(3.12)$ | $0.67_{\pm0.27}(6.63)$ | 3.47 | 2.84 |
| ERM-KTP [23] | $0.00_{\pm0.00}(0.00)$ | $96.62_{\pm0.66}(3.38)$ | $78.10_{\pm0.82}(1.54)$ | $34.27_{\pm0.45}(26.97)$ | 7.97 | 2.02 |
| Boudary [3] | $4.67_{\pm0.11}(4.67)$ | $98.84_{\pm0.97}(1.16)$ | $78.23_{\pm0.69}(1.41)$ | $0.00_{\pm0.00}(7.30)$ | 3.63 | 1.83 |
| GDR-GMA (ours) | $0.00_{\pm0.00}(0.00)$ | $96.26_{\pm0.75}(3.74)$ | $81.76_{\pm0.46}(2.12)$ | $4.00_{\pm0.27}(3.30)$ | **2.29** | 0.75 |

**Comparison experiments in class forgetting.** We also conduct experiments in class forgetting, following the previous works [3, 23, 32]. Except for the baseline methods in random data forgetting and sub-class forgetting, we add two class-wise MU methods as baselines, ERM-KTP [23] and Boundary [3]. As shown in Table 4, GDR-GMA still has the smallest average performance gap and has less time overhead than the SOTA works.

The experimental results in these three scenarios demonstrate the superior effectiveness, fidelity, and efficiency of GDR-GMA. Furthermore, the simple implementation and adaptability across scenarios also show the scalability of GDR-GMA.

**Effectiveness of GDR.** We evaluate the effectiveness of the GDR method in handling the problem of direction conflict. As shown in Fig.8, GDR can significantly alleviate the direction conflict problem. However, Fig. 8(b) shows that the rate of conflict samples is not 0% in the first few epochs due to the approximation error using the gradient bank. Specifically, the approximation error is caused because the cosine similarity between adjacent epochs is around 0.6-0.8, as shown in Fig. 4. Nevertheless, the rate reaches 0% after a few epochs because the error will be negligible when the cosine similarity increases during the unlearning process.

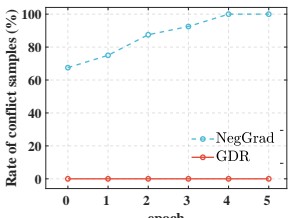
**(a) Direction conflict of $g_f - g_r$**

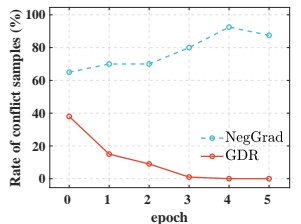
**(b) Direction conflict of $g_r - g_F$**

**Figure 8: Evaluation of the effectiveness of the GDR method on CIFAR-10 using ResNet-18. After GDR rectifies the direction of the conflicting gradients, the rate of conflicting samples is decreased to 0%.**

**Effectiveness of GMA.** Then, we evaluate the effectiveness of the GDR method in handling the gradient dominant problem. As shown in Fig. 9, these two tasks can achieve a balance where each gradient fulfills its duty instead of solely tending to one of them after the GMA method dynamically adjusts the magnitude of the update gradients. Furthermore, it demonstrates that GMA can achieve a balance between the updated gradients.

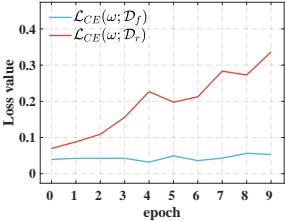

**Figure 9: Evaluation of the effectiveness of the GMA method on CIFAR-10 using ResNet-18. After GMA adjusts the magnitude of the update gradients, the forgetting loss value increases with maintaining the remaining loss.**

## 5.3 Ablation Experiments

In this section, we conduct ablation experiments to further analyze the effectiveness of GDR-GMA.

**Impact of the gradient bank.** In Sec. 4.1, we propose the gradient bank to approximate the set of gradients $\{g_{F_k}\}_{k=1}^{|\mathcal{D}_F|}$ with $g_{joint}$. Fig. 8(b) demonstrates the effectiveness of this approximation using the gradient bank to alleviate the direction conflict. Furthermore, the gradient bank can significantly reduce time overhead without sacrificing performance, as shown in Table 5.

**Table 5: Ablation experiments of the gradient bank on CIFAR-10 using ResNet-18.**

| | $Acc_{\mathcal{D}_f}$ ($\Delta \downarrow$) | $Acc_{\mathcal{D}_r}$ ($\Delta \downarrow$) | $Acc_{val}$ ($\Delta \downarrow$) | MIA ($\Delta \downarrow$) | Avg. Gap $\downarrow$ | RTE $\downarrow$ |
|---|---|---|---|---|---|---|
| **Random Data Forgetting (20%)** | | | | | | |
| w/o Bank | 93.86 (0.14) | 99.23 (0.77) | 92.43 (1.57) | 82.32 (5.56) | 2.01 | 102.32 |
| w/ Bank | 93.84 (0.16) | 99.22 (0.78) | 92.40 (1.60) | 82.36 (5.60) | 2.04 | **1.40** |
| **Random Data Forgetting (50%)** | | | | | | |
| w/o Bank | 92.06 (0.67) | 99.24 (0.76) | 90.40 (1.59) | 77.02 (6.23) | 2.31 | 122.42 |
| w/ Bank | 92.10 (0.63) | 99.07 (0.93) | 90.37 (1.62) | 76.27 (5.48) | 2.17 | **2.12** |

**Impact of $\gamma$ and $\epsilon$.** We propose the dynamic magnitude weight parameter in Eq. 6 to adjust the magnitude of the update gradients, resulting in a balance between the forgetting and remaining tasks. Specifically, we use the steepness parameter $\gamma$ to control how sharply the weight parameter transitions from its minimum value to its maximum value and the constant small value $\epsilon$ to control the deviation of the loss value of the remaining dataset. As shown in Fig. 8, the results show that the value of $\gamma$ does not affect the average performance gap but affects the time overhead. The higher value of $\epsilon$ will reduce the time overhead with an increasing average performance gap.

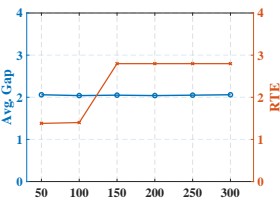
**(a) The steepness parameter $\gamma$**

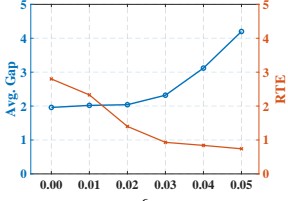
**(b) The constant small value $\epsilon$**

**Figure 10: Ablation experiments of the steepness parameter $\gamma$ (a) and the constant small value $\epsilon$ (b) in Eq. 6.**

## 6 Conclusion

In this paper, we first analyze the problems of the basic gradient ascent MU method in a multi-task learning view and identify two key problems: the gradient direction problem and the gradient dominant problem. To address these problems, we propose the GDR-GMA MU method. For the gradient direction problem, GDR-GMA rectifies the direction of the conflicting gradients. For the gradient dominant problem, GDR-GMA dynamically adjusts the magnitude of the update gradients. Then, we conducted extensive experiments to demonstrate the effectiveness, fidelity, and efficiency of GDR-GMA.

## Acknowledgments

This work was supported in part by the National Natural Science Foundation of China under Grands 61960206014, 62102300, and 62121001, in part by the '111 Center' (B16037), and by the Fundamental Research Funds for the Central Universities (No. YJSJ24010).

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
