# OpenReview forum: "GDR-GMA: Machine Unlearning via Direction-Rectified and Magnitude-Adjusted Gradients"
_acmmm.org/ACMMM/2024/Conference — MM2024 Poster_

### Official Review · Reviewer_KzYJ · 2024-05-21

**Rating:** 3
**Confidence:** 2

**Summary:**

The paper identifies the gradient direction and gradient dominant problem of the basic gradient ascent Machine Unlearning method in a multi-task learning view. And the GDR-GMA are proposed to address above two problems by rectifying the direction of the conflicting gradients and adjusting the magnitude of the update gradients.

**Strengths:**

1. The paper is well written and easy to follow.
2. The authors provide the code.
3. This paper focuses on an important and practical question on sample-wise machine unlearning task : the trade-off between effectiveness, fidelity, and efficiency.
4. The numerical experiments are well executed and show promising performance achieved by the proposed methods.

**Limitations:**

1. The technical contributions of the proposed method are not particularly clear. It seems that the paper directly applies the existing multi-task learning method in machine unlearning scenario. The core techniques of GDR is similar to PCGrad [1] and I believe that other methods tailored for multi-task learning [2, 3, 4, 5] could also be adopted here to address gradient direction and gradient dominant problem.
2. More datasets, especially in different modalities, should be used.

[1] Yu, Tianhe, et al. "Gradient surgery for multi-task learning."\
[2] Wang, Zirui, et al. "Gradient vaccine: Investigating and improving multi-task optimization in massively multilingual models."\
[3] Liu, Bo, et al. "Conflict-averse gradient descent for multi-task learning."\
[4] Liu, Shikun, Edward Johns, and Andrew J. Davison. "End-to-end multi-task learning with attention."\
[5] Guo, Michelle, et al. "Dynamic task prioritization for multitask learning."

**Suitability:**

2

---

### Official Review · Reviewer_pf8J · 2024-05-23

**Rating:** 3
**Confidence:** 3

**Summary:**

This paper introduces a novel machine unlearning method. The authors reveal the gradient direction and gradient domination problem which hinder the machine unlearning algorithm from achieving a good trade-off among effectiveness, fidelity, and efficiency from a multi-task learning view. Therefore, this work proposes the GDR-GMA unlearning method, which rectifies the directions between the conflicting gradients and dynamically adjusts the magnitude of gradients for effectively forgetting data while preserving model utility.
Extensive experiments are conducted on CIFAR-10, CIFAR-20, and Tiny-ImageNet across ResNet-18 and ViTs, and results demonstrate the effectiveness of the proposed unlearning method.

**Strengths:**

- The motivation is clear and the paper is well-written.
- Solving gradient conflicts in machine unlearning from a multi-task learning view is intriguing.
- Experimental results show the proposed method is effective.

**Limitations:**

- Lack of details. For example, it is unclear which classes are chosen to be forgotten in the forgetting scenario (e.g., subclass and class forgetting).
- The setting for baselines seems incorrect. For example, the SalUn method is reported to be trained with a learning rate of 0.1, but the SalUn paper specifies learning rates between 5e-4 and 5e-2. Also, what is the saliency sparsity used for SalUn? I believe it is very important to verify that the baselines are not intentionally set to underperform.
- Additionally, for the proposed method, I am curious why the authors try to compute the gradient per sample, why not just compute the gradient for the mini-batch?

**Suitability:**

2

---

### Official Review · Reviewer_w9ea · 2024-05-25

**Rating:** 4
**Confidence:** 2

**Summary:**

The paper presents an approach for machine unlearning, addressing challenges for effective methods to remove data from AI models without extensive retraining. The authors propose the GDR-GMA method, which includes two key components: Gradient Direction Rectification (GDR) and Gradient Magnitude Adjustment (GMA). GDR solves gradient direction problem by projecting conflicting gradients onto an orthonormal plane, reducing destructive interference, while GMA dynamically adjusts the magnitude of update gradients to balance tasks of forgetting dataset and maintaining the remaining dataset.

**Strengths:**

- The introduction of GDR and GMA methods addresses key aspects of machine unlearning problem. GDR addresses gradient direction conflicts by projecting conflicting gradients onto an orthonormal plane, GMA adjusts magnitude of gradient updates to balance unlearning and maintaining tasks.
- Comprehensive experiments are done across three MU scenarios: random data forgetting, sub-class forgetting, and class forgetting. With comparisons with baseline methods to demonstrate improvements in classification performance, computational efficiency, and the ability to maintain model accuracy.

**Limitations:**

- Experiments are conducted on CIFAR-10, CIFAR-20, and Tiny-ImageNet, which are standard benchmarks in CV. Evaluating the method on a broader range of datasets would provide a more comprehensive understanding of its effectiveness and limitations, e.g. unlearning documents of a NLP dataset with an LLM would be a quite interesting setup.

**Suitability:**

2

---

### Meta-Review · Area_Chair_Vayy · 2024-06-30

**Recommendation:** Accept (Poster)
**Confidence:** 4

**Metareview:**

The reviewers agreed that the motivation is clear and the results are convincing. They also requested that many improvements need to be done.